# The prevalence of undernutrition and associated risk factors in people with tuberculosis in Lao People's Democratic Republic

Donekham Inthavong[1], Hend Elsayed[2]*, Phonesavanh Keonakhone[3], Vilath Seevisay[4], Somdeth Souksanh[4], Sakhone Suthepmany[1], Misouk Chanthavong[1], Xaysomvang Keodavong[1], Phonesavanh Kommanivanh[1], Phitsada Siphanthong[1], Phengsy Sengmany[3], Buahome Sisounon[3], Jacques Sebert[4], Manami Yanagawa[2], Fukushi Morishita[2], Nobuyuki Nishikiori[5], Takuya Yamanaka[5]

1 National Tuberculosis Control Centre, Vientiane, Lao People's Democratic Republic, 2 Integrated Communicable Disease Control, World Health Organization Regional Office for the Western Pacific, Manila, Philippines, 3 National Nutrition Centre, Vientiane, Lao People's Democratic Republic, 4 World Health Organization Representative Office for Lao PDR, Vientiane, Lao People's Democratic Republic, 5 Global Tuberculosis Programme, World Health Organization, Geneva, Switzerland

* hend.elabbasy@hotmail.com

## Abstract

### Objective

Undernutrition is common in individuals with tuberculosis (TB). There is a bidirectional association between having TB and undernutrition; undernutrition increases the risk of having active TB, and having TB worsen undernutrition by reducing appetite and food intake. Despite World Health Organization (WHO) recommendations for comprehensive nutritional assessment and counselling for people with TB, systematic implementation is lacking in Lao People's Democratic Republic (Lao PDR), leading to an insufficient understanding of undernutrition prevalence in this population.

### Methods

A facility-based cross-sectional survey was conducted between March 2022 and March 2023 in six central and provincial hospitals in Lao PDR. We assessed the prevalence of undernutrition in 312 people diagnosed with TB at TB diagnosis using body mass index (BMI). Undernutrition was defined as a BMI < 18.5 kg/m², and severe undernutrition as a BMI below 16.5 kg/m². Data on demographic, clinical and economic information and nutritional status were extracted from an intervention study assessing the effect of nutritional counselling and feeding on the financial burden of TB and TB treatment outcomes.

**Data availability statement:** Study dataset contains privacy-sensitive information including participant's individual and household income that formed a core part of the analysis. Even though we remove patient's identifiers such as patient number and name, there is still a possibility that those who are familiar with the project sites and beneficiaries may be able to identify participants and their households. The informed consent signed by all participants explicitly mentioned that only the research team have access to the data set. Due to such ethical and confidentiality restrictions, the survey dataset will be made available only upon request and with permission from the National Tuberculosis Control Programme (NTP), Ministry of Health, Lao PDR. All interested researchers will contact - the NTP of Lao PDR (ndonekham@gmail.com and phonesa-vanh_33@hotmail.com), and/or - Non-author contact: WHO/WPRO ERC (wproethicsreview-comm@wpro.who.int).

**Funding:** This study was funded by the Government of the Republic of Korea through the Korean Disease Control and Prevention Agency and the Government of Japan through the Ministry of Health, Labour and Welfare. The funders had no role in study design, data collection and analysis, decision to publish, or preparation of the manuscript.

**Competing interests:** The authors have declared that no competing interests exist.

## Results

Of 312 participants, 40.7% (n = 127) were with undernutrition (BMI < 18.5 kg/m$^2$) at the time of TB diagnosis. 20.5% (n = 64) with severe undernutrition (BMI < 16.5 kg/m$^2$). Factors significantly associated with undernutrition included age group 15–24 years (Adjusted odds ratio (AOR) 6.9, 95% confidence interval [95%CI]: 2.2–23.2), drug-resistant TB (AOR 3.2, 95%CI: 1.0–11.8), experiences of hospitalization until TB diagnosis (AOR 3.4, 95%CI: 2.0–5.9), self-reported weight loss (AOR 7.8, 95%CI: 2.3–36.4), and below the poverty line at TB diagnosis (AOR 1.9, 95%CI: 1.0–3.6).

## Conclusion

A high prevalence of undernutrition was observed in people diagnosed with TB at their diagnosis in Lao PDR. The findings underscore the urgent need for systematic nutritional assessment and counselling as integral components of TB care to identify and address undernutrition, thereby enhancing overall health outcomes for individuals with TB.

## Introduction

Undernutrition is a common clinical finding in people with tuberculosis (TB) and a risk factor for developing active TB [1,2]. There is a bidirectional association between having TB and undernutrition. One direction is from undernutrition to active TB: a linear association between body mass index (BMI) and incidence of active TB was reported in a systematic review (one increase of BMI leads to 13.8% reduction in incidence of active TB) [3]. The other direction is from having TB to be with undernutrition: people with TB often experience reduced appetite and weight loss, and become affected by undernutrition due to metabolic changes during TB treatment [3–5]. Given the importance of the management of the nutritional status in people with TB, World Health Organization (WHO) recommends assessing nutritional status for all people with TB and to provide nutritional counselling and care for people with TB with needs and nutritional care and support for people with TB [6,7].

Undernutrition has a negative impact on TB treatment outcomes [8]. Undernutrition is associated also with death due to TB, and a cohort study in India found that low BMI was associated with death during TB treatment [5]. A study found that low BMI was associated with increased all-cause mortality rates in people with drug-resistant TB [9]. A retrospective study conducted in Ethiopia found that people with undernutrition had a higher risk of unsuccessful treatment outcomes [10]. Severe undernutrition risk ranged from 21.3% to 24.9% in people with TB, and baseline undernutrition might be used as a predictor of TB mortality [11]. A recent metanalysis found that people with undernutrition had an increased risk of experiencing unfavourable outcomes (Adjusted Odds Ratio (AOR)=1.7, 95% confidence interval (95%CI): 1.4–1.9), death (AOR = 3.1, 95%CI 2.4–3.9), and failure/recurrence (AOR = 1.6, 95%CI 1.2–2.0) [12]. Another study in India found that undernutrition at the start of treatment and prior to the onset of TB symptoms were linked to adverse outcomes, with adjusted incidence rate ratios (aIRR) of 2.1

(95% CI, 1.4–2.9) and 2.2 (95% CI, 1.2–3.9) respectively. Furthermore, a failure to increase BMI after the commencement of TB treatment was associated with a higher risk of unfavourable treatment outcomes, with an aIRR of 1.8 (95% CI, 1.2–2.6) [13]. These studies suggest that improving BMI at an early stage of TB treatment has an impact on TB treatment outcomes.

TB remains an issue of public health in Lao People's Democratic Republic (Lao PDR) [14]. Although the estimated TB incidence rate is consistently declining at 3.9% per annum since 2000, the rate still remains high at 138 per 100,000 population in 2022, being higher than that of the WHO Western Pacific Region (96 cases per 100,000 population) and the global estimate (133 cases per 100,000 population) [14]. Lao PDR has National Plan of Action on Nutrition 2021–2025, and the national plan includes a strategic objective to prevent TB-related undernutrition [15]. National TB Programme (NTP) in Lao PDR conducted the first national TB patient cost survey in 2018–2019. In the survey, BMI information was available only from people with drug-resistant TB (DR-TB), and the data shows that 33% were affected by severe undernutrition and 28% by moderate undernutrition at the time of TB diagnosis [16]. The results highlighted the necessity to investigate the prevalence of undernutrition and to enhance food/nutritional support for people with TB. However, systematic nutritional assessment and counselling are not provided for people with TB in Lao PDR, and the data on the nutritional status of people with TB such as weight, height and BMI are not well known in the country. Hence, our study aimed at assessing the prevalence of undernutrition in people with TB at their TB diagnosis and also its risk factors to inform policies and multisectoral engagement for TB and undernutrition.

## Methods

### Study setting

Lao PDR has a high burden of TB with an incidence of 138 per 100,000 in 2022 [14]. The NTP conducted a nationwide TB patient cost survey in 2018–2019 using the WHO recommended cross-sectional approach and cost extrapolation method [16]. The survey recommended a wide range of policy intervention to minimize costs incurred by TB affected households. One of the recommendations is to improve nutritional support for people with TB including systematic nutrition assessment, counselling, and therapeutic and supplementary feeding for those in need, in coordination with the national nutrition centre and in line with the national nutrition strategy.

This study is a subset of an ongoing intervention study aiming to measure the effect of nutritional counselling and support on TB treatment outcome and financial burden due to TB (the main study). The main study is being conducted in six central and provincial hospitals with a large number of TB case notifications: Mittaphap hospital and Setthathirath hospital (Vientiane capital), Khammuan provincial hospital (Khammuan province), Luanprabang provincial hospital (Luanprabang province), Savannakhet provincial hospital (Savannakhet province) and Champasack provincial hospital (Champasack province). We collected the data of BMI, self-reported weight and appetite changes, patient costs, income, health service utilization, coping mechanism and social consequences of TB at 4 time points: 1) at time of TB diagnosis and starting TB treatment, 2) at the end of the TB treatment intensive phase, and 3) during the middle and 4) at the end of the TB treatment continuation phase. The sample size of the main study was 314 people with TB, ensuring 80% power to detect a minimum 12.6% reduction in the proportion of TB-affected households who incur catastrophic costs due to TB.

The inclusion criteria of the main study were as follows: 1) participants (including children) who are newly diagnosed as drug-susceptible TB (DS-TB) or drug-resistant TB (DR-TB); 2) consent to the study participation within 7 days after TB diagnosis; 3) including new, relapse and retreatment TB cases.

### Study design and data collection

The enrolment data, which were collected at the time of TB diagnosis, were derived from the main study. The data analysed in this study were obtained from all of the hospitals included in the main study. The collection of the enrolment data was conducted between 13 March 2022–March 2023.

Clinical dieticians stationed at each study site served as interviewers for participants recruitment. Prior to being deployed to each study site, clinical dieticians received a 5-day training course on the study objectives, methods, and data collection tools. In addition to that, we conducted a 5-day pilot data collection to familiarize themselves with the data collection process. During participants enrolment, clinical dieticians explained the purpose of the main study and shared a written information sheet, in relevant local languages. Those who agreed to participate in the research and signed the informed consent form were enrolled. For minor participants (aged below 18), written informed consent was obtained from their parents or guardians. The data collection at the time of TB diagnosis was conducted via in-person interviews at study sites, and the interview time was around 30–45 minutes.

The collected data included: 1) demographic information (age, sex, education, marital status household size, employment status, insurance status), 2) clinical information (mode of TB diagnosis, previous TB history, HIV status, smoking/alcohol behaviour, drug use, other comorbidities), 3) nutritional assessment (weight, height, BMI, current appetite), 4) TB patient costs, household assets and income, coping mechanism, perceived financial impact before TB diagnosis.

Anthropometric data were measured using standardized weight and height scales.

Costs consist of direct medical costs (e.g., medical consultation fees, costs for drugs and hospitalization costs) and direct non-medical costs (e.g., costs for transportation, food and supplements, and accommodation incurred for health facility visits) as well as indirect costs (i.e., income loss) during care seeking for TB symptoms [17]. We adapted the data collection tools used in the Lao PDR national TB patient cost survey, originally designed as a cross-sectional study based on the WHO handbook for national TB patient cost surveys, into a longitudinal study design [16,18–20].

## Data analysis

Data were collected and entered at the time of interviews using tablet-based questionnaires with Ona and Open Data Kit (ODK) collect. Data cleaning and processing, statistical analyses, and data visualizations were performed using R-4.2.0 software (CRAN: Comprehensive R Archive Network).

For continuous data, descriptive statistics included mean with standard deviation (SD) and 95% CI, and median with interquartile range (IQR). Categorical data were presented as frequencies with proportion (%). A cut-off of BMI 18.5 kg/$m^2$ was used to define undernutrition at the time of TB diagnosis (BMI < 18.5 kg/$m^2$ categorized as undernutrition) [21,22]. Individuals are classified as living below the poverty line if their daily income is less than 1.9 US dollars PPP (international poverty line) [23]. Statistical differences between people with and without a low BMI were tested using a chi-square test for categorical data and either the t-test or Kruskal–Wallis test for continuous data. Statistical significance was defined as a p-value less than 0.05. Costs and incomes were initially recorded in Lao Kip (LAK) and later converted into United States dollars (US$) for analysis at the rate of LAK 17,531.12 to US$ 1 using the average UN Operational Rates of Exchange during the data collection period (March 2022-March 2023).

Univariate logistic regression analysis was conducted to identify variables associated with undernutrition at TB diagnosis. Multivariate backward stepwise logistic regression was performed to identify the best model based on the Akaike information criterion (AIC). The selected final model was used for multivariate logistic regression analysis to calculate AOR and 95% CI.

## Ethical considerations

A written consent form was obtained from each participant prior to enrolment. The form underscored the protection of participant's rights and confidentiality, explicitly stating that only the study investigators would have access to the study dataset. Prior to obtaining a written informed consent, our data collectors explained the purpose of this research using a written information sheet during participant's waiting time at each study site. Ethical clearances were obtained from the Lao PDR National Ethical Committee and National Institute of Public Health (reference number 021/ NECHR) and the WHO Research Ethics Review Committee in the Western Pacific Region (reference number 2022.3.LAO.1.ETB).

Additionally, we obtained permission/endorsement from NTP and the department of disease control of Lao PDR Ministry of Health as well as from the hospital directors of study sites, to conduct the data collection. All methods were performed following the relevant guidelines and regulations established by the research approving institutions.

## Results

### Characteristics of study population

A total of 312 people with TB were enrolled in the study. The demographic characteristic of the total study participants is shown in **Table 1**. 62.5% male while 37.5% are female. 20.8% were aged 65 or older. 60.6% were married while 24.4% were single. Only 19.2% were highly educated at college or higher level. 25.3% were unemployed and 46.8% engaged in informal paid work. National health insurance (NHI) covered 54.2% of participants while 37.2% reported no health insurance. Notably, 50.3% never smoked, and 72.4% never or rarely drunk alcohol. The majority of participants were on DS-TB treatment while 5.1% were on MDR-TB treatment. The proportion of people living with HIV was 13.8%. Most of participants (93.6%) were new cases with no previous history of TB treatment. Nearly half of participants (49.4%) faced a diagnostic delay with more than four weeks from onset of TB symptoms. Additionally, 34.9% were admitted to the hospital until the confirmation of the TB diagnosis.

### Nutritional status

Of 312 participants, 127 (40.7%) were identified with undernutrition (BMI < 18.5 kg/m$^2$), comprising 20.2% moderate under-nutrition (BMI between 16.5 and 18.5 kg/m$^2$) and 20.5% severe undernutrition (BMI < 16.5 kg/m$^2$) while 59.3% were nour-ished (BMI: ≥ 18.5 kg/m$^2$).

### Demographic characteristics

The proportion of people with TB aged 25–34 years among people with BMI < 18.5 kg/m$^2$ (21.3%) was significantly higher than that among people with BMI > 18.5 kg/m$^2$ (15.7%). Similarly, the proportions of those aged 65 years and above with BMI < 18.5 kg/m$^2$ (24.4%) and those aged 15–24 years with BMI < 18.5 kg/m$^2$ (15%) were notably higher than those in the same age groups among the BMI > 18.5 kg/m$^2$ (18.4% and 4.9%, respectively, p = 0.003).

Regarding marital status, the proportion of single amongst people with undernutrition was significantly higher (32.3%) compared to nourished (18.9%) with P = 0.026.

In terms of insurance coverage, the distribution of non-insured participants was almost consistent across both BMI categories. The predominant insurance type was NHI, encompassing 56.7% of people with undernutrition and 52.4% of those nourished. However, no statistically significant difference was observed (p-value = 0.819).

It is worth noting that amongst people with TB, those with undernutrition had a higher proportion of current smokers (13.4%) compared to nourished (8.1%). However, this difference did not reach statistical significance with a p-value of 0.319. Similarly, the proportion of daily drinkers was slightly higher among undernutrition individuals (10.2%) compared to nourished (6.5%), though the difference was not statistically significant (p-value = 0.407).

Regarding education levels, both the proportion of uneducated (12.6%,) or having only primary education (29.9%) among undernutrition individuals was greater than among well-nourished individuals (8.6% and 24.3%, respectively), yet none of these differences reached statistical significance (p = 0.373). Lastly, unemployment and retirement rates were also higher among individuals with undernutrition (unemployment: 28.3%; retirement: 16.5%) when compared to well-nourished individuals (unem-ployment: 23.2%; retirement: 9.7%), although this difference was not statistically significant (p-value of 0.088).

### Clinical characteristics

The proportion of multidrug-resistant/rifampicin-resistant (MDR/RR-TB) amongst people with undernutrition (8.7%) was significantly higher than that among nourished people (2.7%) with p = 0.037. The percentage of people living with

**Table 1. Demographic and clinical characteristics of study participants, by body mass index of participants.**

| Variable | Category | Total | with BMI < 18.5 kg/m² | with BMI ≥ 18.5 kg/m² | p-value |
|---|---|---|---|---|---|
| | | N (%) | N (%) | N (%) | |
| **Total** | | **312 (100%)** | **127 (40.7%)** | **185 (59.3%)** | **–** |
| **Nutritional status** | | | | | |
| BMI | Mean (SD) | 19.6 (3.6%) | 16.3 (1.4%) | 22.0 (2.7%) | <0.001 |
| | BMI: < 16.5 | 64 (20.5%) | 64 (50.4%) | 0 (0.0%) | <0.001 |
| | BMI: 16.5–18.5 | 63 (20.2%) | 63 (49.6%) | 0 (0.0%) | |
| | BMI: ≥ 18.5 | 185 (59.3%) | 0 (0.0%) | 185 (100.0%) | |
| **Demographic characteristics** | | | | | |
| Age group | 0–14 | 1 (0.3%) | 0 (0.0%) | 1 (0.5%) | **0.003** |
| | 15–24 | 28 (9.0%) | 19 (15.0%) | 9 (4.9%) | |
| | 25–34 | 56 (17.9%) | 27 (21.3%) | 29 (15.7%) | |
| | 35–44 | 42 (13.5%) | 14 (11.0%) | 28 (15.1%) | |
| | 45–54 | 58 (18.6%) | 14 (11.0%) | 44 (23.8%) | |
| | 55–64 | 62 (19.9%) | 22 (17.3%) | 40 (21.6%) | |
| | 65+ | 65 (20.8%) | 31 (24.4%) | 34 (18.4%) | |
| Sex | Female | 117 (37.5%) | 47 (37.0%) | 70 (37.8%) | 0.976 |
| | Male | 195 (62.5%) | 80 (63.0%) | 115 (62.2%) | |
| Marital status | Single | 76 (24.4%) | 41 (32.3%) | 35 (18.9%) | **0.026** |
| | Married | 189 (60.6%) | 69 (54.3%) | 120 (64.9%) | |
| | Divorced/separated/widowed | 47 (15.1%) | 17 (13.4%) | 30 (16.2%) | |
| Insurance type | None | 116 (37.2%) | 46 (36.2%) | 70 (37.8%) | 0.819 |
| | National Health Insurance (NHI) scheme | 169 (54.2%) | 72 (56.7%) | 97 (52.4%) | |
| | Community-Based Health Insurance (CBHI) | 3 (1.0%) | 2 (1.6%) | 1 (0.5%) | |
| | Health Equity Fund (HEF) | 0 (0.0%) | 0 (0.0%) | 0 (0.0%) | |
| | Social Security Organization (SSO) for salaried private-sector employees | 9 (2.9%) | 3 (2.4%) | 6 (3.2%) | |
| | State Authority for Social Security (SASS) for civil servants | 11 (3.5%) | 3 (2.4%) | 8 (4.3%) | |
| | Private health insurance | 3 (1.0%) | 1 (0.8%) | 2 (1.1%) | |
| | Other | 1 (0.3%) | 0 (0.0%) | 1 (0.5%) | |
| Smoker | No smoking experience | 157 (50.3%) | 62 (48.8%) | 95 (51.4%) | 0.319 |
| | Current smoker | 32 (10.3%) | 17 (13.4%) | 15 (8.1%) | |
| | Ex-smoker | 123 (39.4%) | 48 (37.8%) | 75 (40.5%) | |
| Alcohol use | Daily | 25 (8.0%) | 13 (10.2%) | 12 (6.5%) | 0.407 |
| | Weekly | 31 (9.9%) | 12 (9.4%) | 19 (10.3%) | |
| | Monthly | 30 (9.6%) | 15 (11.8%) | 15 (8.1%) | |
| | Rarely/Never | 226 (72.4%) | 87 (68.5%) | 139 (75.1%) | |
| Educational level | No education | 32 (10.3%) | 16 (12.6%) | 16 (8.6%) | 0.373 |
| | Primary | 83 (26.6%) | 38 (29.9%) | 45 (24.3%) | |
| | Lower/higher secondary | 137 (43.9%) | 51 (40.2%) | 86 (46.5%) | |
| | Diploma or higher, vocational, other | 60 (19.2%) | 22 (17.3%) | 38 (20.5%) | |
| Employment status before TB | Unemployed | 79 (25.3%) | 36 (28.3%) | 43 (23.2%) | **0.088** |
| | Formal paid work | 48 (15.4%) | 14 (11.0%) | 34 (18.4%) | |
| | Informal paid work | 146 (46.8%) | 56 (44.1%) | 90 (48.6%) | |
| | Retired/student/housework/other | 39 (12.5%) | 21 (16.5%) | 18 (9.7%) | |

*(Continued)*

**Table 1.** (Continued)

| Variable | Category | Total | with BMI < 18.5 kg/m² | with BMI ≥ 18.5 kg/m² | p-value |
|---|---|---|---|---|---|
| | | N (%) | N (%) | N (%) | |
| **Clinical characteristics** | | | | | |
| Site and diagnosis of TB | Pulmonary, bacteriologically confirmed | 225 (72.1%) | 91 (71.7%) | 134 (72.4%) | 0.841 |
| | Pulmonary, bacteriologically unconfirmed (clinically diagnosed) | 70 (22.4%) | 30 (23.6%) | 40 (21.6%) | |
| | Extrapulmonary | 17 (5.4%) | 6 (4.7%) | 11 (5.9%) | |
| Drug resistance status | TB (first-line treatment) | 296 (94.9%) | 116 (91.3%) | 180 (97.3%) | **0.037** |
| | MDR/RR-TB | 16 (5.1%) | 11 (8.7%) | 5 (2.7%) | |
| Treatment history | New | 292 (93.6%) | 121 (95.3%) | 171 (92.4%) | 0.174 |
| | Relapse | 15 (4.8%) | 3 (2.4%) | 12 (6.5%) | |
| | Retreatment | 5 (1.6%) | 3 (2.4%) | 2 (1.1%) | |
| | Other | 0 (0.0%) | 0 (0.0%) | 0 (0.0%) | |
| HIV status | HIV negative | 268 (85.9%) | 101 (79.5%) | 167 (90.3%) | **0.019** |
| | HIV positive | 43 (13.8%) | 25 (19.7%) | 18 (9.7%) | |
| | HIV test not done | 1 (0.3%) | 1 (0.8%) | 0 (0.0%) | |
| Diagnostic delay (>4 weeks from onset of TB symptoms) | No | 158 (50.6%) | 60 (47.2%) | 98 (53.0%) | 0.379 |
| | Yes | 154 (49.4%) | 67 (52.8%) | 87 (47.0%) | |
| Hospitalization during care seeking | Not hospitalized | 203 (65.1%) | 60 (47.2%) | 143 (77.3%) | **<0.001** |
| | Hospitalized | 109 (34.9%) | 67 (52.8%) | 42 (22.7%) | |

HIV among those with BMI < 18.5 kg/m² was 19.7% which was significantly higher than 9.7% among participants with BMI > 18.5 kg/m² (p = 0.019). The proportion of hospitalization until TB diagnosis was significantly higher among individuals with BMI < 18.5 kg/m² (52.8%) compared to those people with BMI > 18.5 kg/m² (22.7%) (p < 0.011)

## Self-reported weight changes and appetite

A 90.1% of the study participants reported a weight loss in the last 3–6 months, and the proportion was higher among those with undernutrition (97.6%) compared to nourished participants (84.9%) with p = 0.001 (**Table 2**). There were 80.3% of participants with undernutrition who reported a poor appetite, which was significantly higher (p = 0.006) than among people with TB with BMI > 18.5 kg/m² (65.4%). A higher proportion of severe decrease in food intake was reported from people with undernutrition, i.e., BMI < 16.5 kg/m² (32.3%) compared to nourished people (15.7%, p = 0.002).

## Self-reported income, impoverishment, health service utilization

The overall mean of self-reported monthly household income was US$ 410 before having TB symptoms, which reduced to US$ 338 at the time of TB diagnosis (**Table 3**). The participants with undernutrition faced a larger loss of household income by 31.0% (US$ 131), from US$ 424 before TB diagnosis to US$ 293 at TB diagnosis; compared to those who were nourished, from US$ 399 before being diagnosed with TB to US$ 369 at TB diagnosis, though no statistical significance was observed in the mean household income at the time of TB diagnosis between those with BMI < 18.5 kg/m² and BMI ≥ 18.5 kg/m² (p = 0.249).

In line with the changes in monthly household income, the proportion of households living under poverty at the time of TB diagnosis was notably greater among participants with BMI < 18.5 kg/m² standing at 24.0%, compared to those with BMI ≥ 18.5 kg/m², where it was 11.0% (p = 0.004).

**Table 2. Changes in body weight and appetite, by body mass index of participants.**

| Variable | Category | With BMI < 18.5 kg/m² | With BMI ≥ 18.5 kg/m² | Total | p-value |
|---|---|---|---|---|---|
| | | N (%) | N (%) | N (%) | |
| Weight changes last 3–6 months | Weight stayed same | 3 (2.4%) | 26 (14.1%) | 29 (9.3%) | **0.001** |
| | Weight increase | 0 (0.0%) | 2 (1.1%) | 2 (0.6%) | |
| | Weight decrease | 124 (97.6%) | 157 (84.9%) | 281 (90.1%) | |
| Current appetite | Good appetite | 25 (19.7%) | 64 (34.6%) | 89 (28.5%) | **0.006** |
| | Poor appetite | 102 (80.3%) | 121 (65.4%) | 223 (71.5%) | |
| Appetite last month | Increase food intake | 6 (4.7%) | 5 (2.7%) | 11 (3.5%) | **0.002** |
| | No decrease food intake | 23 (18.1%) | 56 (30.3%) | 79 (25.3%) | |
| | Moderate decrease food intake | 57 (44.9%) | 95 (51.4%) | 152 (48.7%) | |
| | Severe decrease food intake | 41 (32.3%) | 29 (15.7%) | 70 (22.4%) | |

The mean total costs incurred up to the point of TB diagnosis for participants with undernutrition was US$ 640, which was significantly higher than the mean cost for nourished participants (US $183) with p-value of 0.032. The total costs were driven by income loss, as a form of indirect costs. The income loss in participants with BMI < 16.5 kg/m² was US$ 443 (69.0% of the total costs), which was higher than those with BMI > 18.5 kg/m² (US$ 48, p = 0.061). Additionally, hours lost for health facility visits until TB diagnosis was notably greater for individuals with BMI < 18.5 kg/m² (28 hours) compared to those with BMI ≥ 18.5 kg/m² (13 hours), with p-value of <0.001.

Overall, 17.6% of study participants used their savings, borrowed money or sold their assets to cope with financial burden due to TB (Table 4). More than half of study participants faced social consequences of TB including food insecurity,

**Table 3. Changes in monthly reported income, incurred costs, and health service utilization until TB diagnosis.**

| Variable | | With BMI < 18.5 kg/m² | With BMI ≥ 18.5 kg/m² | Total | p-value |
|---|---|---|---|---|---|
| **Monthly self-reported patient's income** | | Mean (SD) | Mean (SD) | Mean (SD) | |
| Patient's income before having TB symptoms | | 135 (318) | 161 (389) | 150 (362) | 0.528 |
| Patient's income at the TB diagnosis | | 74 (279) | 128 (368) | 106 (335) | 0.167 |
| **Monthly self-reported household's income** | | Mean (SD) | Mean (SD) | Mean (SD) | |
| Household income before having TB symptoms | | 424 (743) | 399 (641) | 410 (683) | 0.751 |
| Monthly self-reported Household income at the TB diagnosis | | 293 (470) | 369 (629) | 338 (570) | 0.249 |
| **Impoverishment: TB-affected households below poverty line** | | N (%) | N (%) | N (%) | |
| Before having TB symptoms | | 14 (11%) | 11 (6%) | 25 (8%) | 0.158 |
| At the time of TB diagnosis | | 30 (24%) | 20 (11%) | 50 (16%) | **0.004** |
| **Cost incurred until TB diagnosis, output approach** | | Mean (SD) | Mean (SD) | Mean (SD) | |
| Direct medical costs | | 68 (117) | 64 (145) | 65 (134) | 0.795 |
| Direct non-medical costs | Nutrition & supplement costs | 33 (49) | 23 (58) | 27 (55) | 0.138 |
| | Other nonmedical costs (travel, food and accommodation costs during facility visits) | 97 (453) | 47 (136) | 67 (307) | 0.166 |
| | Total | 129 (470) | 71 (150) | 95 (322) | 0.115 |
| Total direct costs | | 197 (510) | 135 (235) | 160 (373) | 0.146 |
| Indirect costs (Income loss) | | 443 (2820) | 48 (167) | 209 (1810) | 0.061 |
| Total costs until TB diagnosis | | 640 (2870) | 183 (312) | 369 (1856) | 0.032 |
| **Health service utilization** | | Mean (SD) | Mean (SD) | Mean (SD) | |
| Number of facilities visits until TB diagnosis | | 2 (1) | 2 (2) | 2 (2) | 0.336 |
| Number of hours spent for facility visits until TB diagnosis | | 28 (36) | 14 (26) | 20 (31) | <0.001 |

**Table 4. Coping mechanism and social consequences of TB until TB diagnosis.**

| Variable | Category | With BMI < 18.5 kg/m² | With BMI ≥ 18.5 kg/m² | Total | p-value |
|---|---|---|---|---|---|
| | | N (%) | N (%) | N (%) | |
| Coping mechanism | Dissaving | 10 (7.9%) | 20 (10.8%) | 30 (9.6%) | 0.426 |
| | Taking loans | 9 (7.1%) | 5 (2.7%) | 14 (4.5%) | 0.183 |
| | Selling household assets | 6 (4.7%) | 5 (2.7%) | 11 (3.5%) | 0.523 |
| | Any of above | 25 (19.6%) | 30 (16.2%) | 55 (17.6%) | 0.541 |
| Social consequences | Food insecurity | 2 (1.6%) | 2 (1.1%) | 4 (1.3%) | 1.000 |
| | Divorce/separation | 1 (0.8%) | 3 (1.6%) | 4 (1.3%) | 0.896 |
| | Loss of Income (not permanent job loss) | 32 (25.2%) | 36 (19.5%) | 68 (21.8%) | 0.286 |
| | Loss of Job | 29 (22.8%) | 28 (15.1%) | 57 (18.3%) | 0.114 |
| | Interrupted schooling | 2 (1.6%) | 1 (0.5%) | 3 (1.0%) | 0.742 |
| | Social exclusion | 21 (16.5%) | 29 (15.7%) | 50 (16.0%) | 0.963 |
| | Any of above | 72 (56.7%) | 93 (50.3%) | 165 (52.9%) | 0.317 |
| Perceived financial impact of TB | Richer | 0 (0.0%) | 0 (0.0%) | 0 (0.0%) | 0.088 |
| | Unchanged | 78 (61.4%) | 134 (72.4%) | 212 (67.9%) | |
| | Poorer | 44 (34.6%) | 48 (25.9%) | 92 (29.5%) | |
| | Much poorer | 5 (3.9%) | 3 (1.6%) | 8 (2.6%) | |

divorce, income loss, job loss, interrupted schooling or social exclusion. Income loss (21.8%) and job loss (18.3%) were the most common consequences of TB. No statistically significant differences in coping mechanisms and social consequences of TB were observed between those with BMI < 18.5 kg/m² and BMI ≥ 18.5 kg/m².

Perceived financial impact of TB disease episode was categorized into four groups; richer, unchanged, poorer and much poorer. There was no statistically significant difference in the financial impact perception between the two BMI groups (p-value = 0.088). Notably, a higher percentage of people with TB with BMI < 18.5 kg/m² report being poorer and much poorer (38.5%) compared to those who were with BMI ≥ 18.5 kg/m² (27.5%).

The results of a multivariable analysis adjusted for age group, drug resistance status, hospitalization until TB diagnosis, weight loss, and below international poverty line are presented in **Fig 1** (results with univariate and multivariate analysis are available in S1 Table in appendix). There were substantially heightened risk of low BMI among different age groups. Specifically, individuals aged 15–24 years old exhibited a remarkably higher risk (AOR: 6.9, 95%CI: 2.2–23.2) compared to those aged 35–44 years old. The presence of drug resistance also emerged as a significant risk factor for low BMI (AOR: 3.2, 95% CI = 1.0–11.8). Moreover, participants who were hospitalized before the diagnosis of TB faced a substantially higher risk of low BMI (AOR: 3.4, 95% CI = 2.0–5.9). This indicates that the severity of the TB condition, requiring hospitalization, is associated with a greater likelihood of low BMI at the time of TB diagnosis. Notably, individuals who experienced weight loss in the last 6 months prior to TB diagnosis were at a significantly elevated risk of having a low BMI (AOR: 7.8, 95% CI: 2.3–36.4). Finally, participants who were categorized as being below the international poverty line at the time of TB diagnosis also faced an increased risk of low BMI (AOR: 1.9, 95% CI 1.0–3.6).

## Discussion

### The prevalence of undernutrition

Our study assessed the prevalence of undernutrition in people with TB at the time of their TB diagnosis in Lao PDR. The prevalence of undernutrition (BMI < 18.5 kg/m²) in this study was high at 40.7%, and furthermore, severe undernutrition (BMI < 16.5 kg/m²) was also common (20.5%) in our study, which was much higher than the prevalence of undernourishment among the general population of Lao PDR in 2021 (5%) [24]. This finding was consistent with several studies

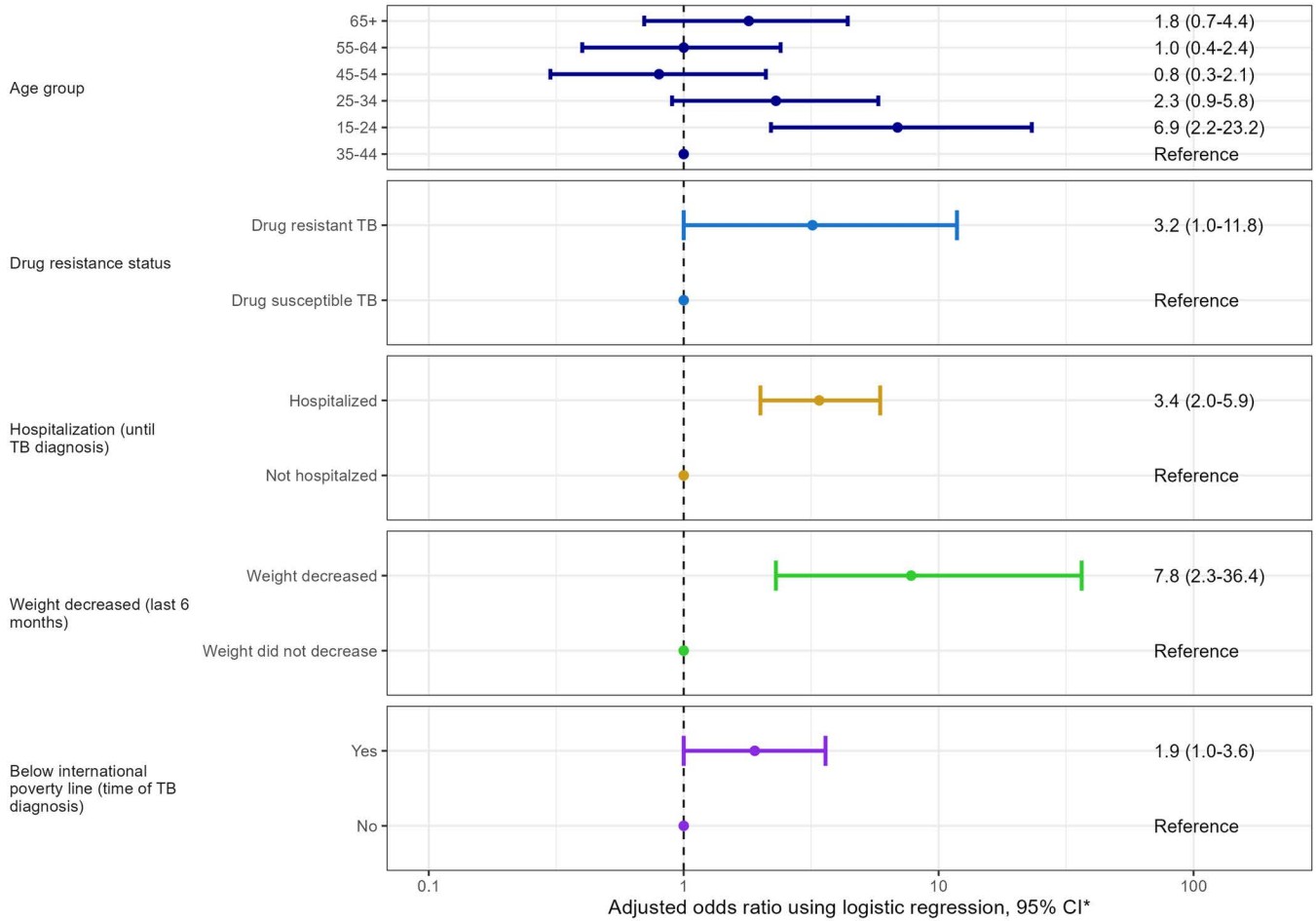

**Fig 1. Risk factors associated with a low body mass index at the time of TB diagnosis.** *x-axis is with log-scale.

conducted in Ethiopia (40.0%-46.0%) [25–27] and Kenya (43.0%) [28]. A higher prevalence of undernutrition among people with TB was reported in other studies; 46.0% in Uganda [29], 57.0% of adult with TB in Malawi [30], 51.0% in Ghana [31], 57.0% in Ethiopia [32], 78.0% in Cameron [33], and 48.0% in a recent systematic review and metanalysis study were undernutrition [34]. A high prevalence of undernutrition among people with TB has been reported also in Asia Pacific countries including 66.0% in India [35], 87.0% in Indonesia [36], and all people newly diagnosed with TB at baseline in Brazil [37].

Studies have shown that the prevalence of undernutrition among people with TB varies depending on the study and the population being studied. However, the difference in the prevalence of undernutrition among people with TB may be affected by survey methods. In addition to the differences in socioeconomic and demographic factors, lifestyle, dietary patterns, the severity of the disease, and the sample size [26,38,39]. Also, some studies were not nationally representative, they were based on data from a single hospital or health centre.

Also, the prevalence of undernutrition among people with TB can vary significantly based on the timing of the studies and the methodology implemented to evaluate their nutritional status. Some studies have chosen to measure BMI at the point of TB diagnosis, before starting TB medication. This approach can potentially exaggerate the extent of undernutrition, primarily because people with TB are often in a critical state of undernutrition during this phase [40]. In contrast, other

investigations have opted to assess BMI after finishing TB treatment, once participants have already recovered from both TB and undernutrition. This alternative approach may inadvertently lead to an underestimation of the degree of undernutrition within this population [41]. In addition to that, the inclusion criteria of study subjects can affect the prevalence of undernutrition among people with TB. For instance, some studies included only adult people with TB while other studies included only pulmonary tuberculosis (PTB) [42]. Also, some studies included pregnant people with TB which might increase or underestimate the level of undernutrition among people with TB [43].

**Risk factors associated with undernutrition**

This study observed a notable discrepancy in the prevalence of undernutrition among people with TB aged between 15–24 years. They had a seven-fold higher likelihood of undernutrition compared to their counterparts aged 35–44 years. This observed pattern may stem from the relative inexperience of younger individuals (aged 15–24 years) in effectively navigating the complexities of the healthcare system and accessing the available medical services [44,45]. Furthermore, their dietary practices were suboptimal, including a tendency to dine away from home and occasional meal skipping. This occurred despite the heightened nutritional requirements necessitated by their ongoing growth and development [46]. Additionally, the younger population is more susceptible to substance use disorders (e.g., alcohol, smoking, and IV drug substances), infectious diseases like HIV, and generally possesses limited awareness regarding nutrition and dietary intake [47–49]. A combination of these factors might contribute to elevating the risk of undernutrition among people with TB in this age group.

In our study, the odds ratio having undernutrition was three times higher among people with MDR/RR-TB compared with DS-TB. This was consistent with a study conducted in Guinea which found that people with MDR/RR-TB were more likely to experience undernutrition than those with DS-TB [50]. Besides, the adverse effects of medications are more pronounced in the treatment of DR-TB as opposed to DS-TB [51]. Nausea, vomiting, or alterations in taste can significantly impact a patient's willingness and ability to intake food [51]. Undernutrition can weaken the immune system which increases the risk of developing MDR-TB and worsens the prognosis for those who already have it [50]. Undernutrition is associated with a higher risk of death and poorer treatment outcomes in people with MDR/RR-TB [52]. In Europe, poor nutrition in people with MDR/RR-TB leads to longer recovery times, worse treatment outcomes, and higher mortality rates [8]. A comprehensive review of 43 studies involving 21,266 people with MDR/RR-TB found that those with undernutrition were twice as likely to experience unsuccessful treatment outcomes and three times higher risk of mortality [53]. Similarly, a study in Indonesia showed that people with MDR/RR-TB who suffered from undernutrition had a 1.9 times higher mortality rate compared to people with MDR/RR-TB with normal nutritional status [54]. These findings underscore the critical need for nutritional counselling and interventions among people with MDR/RR-TB. Given that DR-TB is often managed in larger hospitals with existing nutrition services, integrating nutritional support into DR-TB care could be a vital step, especially in resource-limited settings.

The odds ratio having undernutrition was 3.4 times higher in those who experienced hospitalizations during care seeking for TB, compared to those who were not hospitalized. This can be explained by several factors. 1) **Impact of hospitalization**: Hospitalized people due to TB might be more likely to experience undernutrition due to the impact of hospitalization on their nutritional status. Hospitalization can affect their nutritional status in various ways including: Firstly, hospitalization can trigger a reduced appetite stemming from the hospital environment and stress-related factors like anxiety and depression, both of which can lead to undernutrition and weight loss [39,55]. Secondly, hospitalization can lead to a change in dietary habits, which can result in a lack of essential nutrients [55]. Thirdly, hospitalization can lead to the administration of drugs that can cause gastrointestinal disturbances, such as nausea and vomiting, which can affect the patient's willingness and ability to consume food [32]. Fourthly, hospitalization can lead to a lack of access to food, particularly in low-income settings, which can result in undernutrition [8]. Finally, neglecting nutritional care during hospitalization, providing insufficient meal options, or delaying the addressing of dietary needs within hospital practices and policies

could lead to a decline in nutritional status [56]. A study conducted in Dutch hospital found that 30% of people admitted well-nourished had malnutrition during their stay, with 82% remaining with undernutrition throughout their hospitalization [57]; 2) **Poor nutritional status before hospitalization**: A study conducted in Vietnam found that people with pulmonary TB had poor nutritional status before hospitalization, with 45.8% of people classified as undernutrition based on BMI. This suggests that people with TB may already be at risk of undernutrition before hospitalization, which may be increased during hospitalization [58]. 3) **Impact of undernutrition on TB prognosis**: undernutrition is associated with a higher risk of death and poorer treatment outcomes in people with TB. People hospitalized due to TB may be more likely to experience undernutrition, which can worsen their prognosis and increase their risk of mortality [21]. 4) **Impact of TB on nutritional status**: TB can contribute to undernutrition by causing loss of appetite, weight loss, and nutrient malabsorption. People hospitalized due to TB may be more likely to experience these symptoms, which can lead to undernutrition.

People who experienced weight loss before being diagnosed with TB were nearly 7.8 times more likely to become affected by undernutrition compared to those who did not report weight loss. This suggests that pre-existing undernutrition may contribute to both the development of TB and the weight loss experienced by participants. This was consistent with a systematic review study which confirmed that Individuals with undernutrition are more likely to develop active TB compared to those with a healthy body weight [59].

The proportion of those living under the international poverty line was 8% before having TB symptoms and 16% at the time of TB diagnosis in our study population, which was higher than that in the general population in Lao PDR (7.1%). We also found that the people living below the international poverty line were almost two times more likely to develop undernutrition. This was consistent with a previous study that was conducted in West Bengal, India which suggests that poverty may contribute to undernutrition in people with TB which may lead to unsuccessful treatment outcomes [60]. People living in poverty were facing food insecurity. Food insecurity results in undernutrition which is associated with a higher risk of TB disease and poor treatment outcomes [61]. A matched case-control study of people with pulmonary TB in Ethiopia found that household food shortage was an independent risk factor for active pulmonary TB [61]. This suggests that poverty and food insecurity may contribute to undernutrition in people with TB. Poverty facilitates the transmission of Mycobacterium tuberculosis, primarily through its influence on living conditions, such as people living in overcrowded and poorly ventilated homes, prolonged diagnostic delay, and increased vulnerability due to undernutrition and/or HIV infection [62]. This ecological association suggests that poverty may contribute to undernutrition in people with TB. TB is strongly influenced by social and economic development and health-related risk factors such as undernutrition [63]. Globally in 2020, an estimated 1.9 million incident cases of TB were attributable to undernutrition [63]. A study conducted in India found that the prevalence of TB is significantly higher among the multidimensional poor population compared to the multidimensional non-poor population [64].

Finally, this study proved that TB-affected households with a patient BMI < 18.5 kg/m$^2$, experienced significant economic hardships characterized by income reduction prior to TB diagnosis. These financial challenges may stem from the severity of TB symptoms such as poor appetite and weight loss, as well as a decline in both moral and physical strength, particularly impacting individuals in their prime working years. It's noteworthy that a significant portion of impoverished Laotian individuals engage in physically demanding manual labour in sectors such as agriculture, forestry, construction, and factories. or the costs associated with hospitalization during the pursuit of medical care. Consequently, individuals suffering from undernutrition appear to be at a heightened risk of financial vulnerability within the context of Lao PDR. Therefore, it is imperative to prioritize the expansion of current social and financial support mechanisms to aid this particular population. In Lao PDR, current social and financial support mechanisms for people with TB include free TB services provided in public health facilities, the NHI scheme, which covers 79.3% of the population as of 2019 [65]. While continuous efforts are needed to expand the coverage of the NHI and to ensure that at least direct costs related to TB and nutritional care services are covered, there is an urgent need for established mechanisms to protect TB-affected households from indirect costs – a major driver of catastrophic costs due to TB. This is particularly crucial for informal paid workers who were

vulnerable and socio-economically unstable during TB treatment. Implementing mechanisms such as income replacement schemes or cash transfers can help mitigate the loss of income experienced by people with TB, especially those in the informal sector who lack job security and access to sick leave benefits. Additionally, providing nutritional counselling and care at the early stages may represent a cost-effective approach to pre-empting catastrophic costs before they escalate significantly.

## The current policy for BMI assessments

The current policy for providing BMI assessment for people with TB is not consistent across different regions and countries. In some areas, BMI is not being calculated at the programmatic level [66–68]. In Lao PDR, there are no guidelines to provide BMI assessment for all people with TB. The Lao PDR National TB patient cost survey conducted in 2018–2019 collected patient information from TB treatment cards, the BMI data were available only from people DR-TB [16]. Additionally, a study conducted in Lao PDR to assess the risk of latent tuberculosis infection in children living in households with people with TB assessed the patient's nutritional status by calculating BMI and Middle Upper Arm Circumference (MUAC) [69]. The study found that the prevalence of severe chronic undernutrition in children was 74% which has been reported from a study in the Luangnamtha province [69,70].

Previous studies have proposed using BMI as a method for assessing nutritional status and weight gain among people with TB [66,71]. BMI can also be used to classify the severity of undernutrition at diagnosis and ascertain nutritional recovery. Given the high prevalence of undernutrition in our study, improved policies are required to ensure that BMI assessment is consistently performed for people with TB. This would aid in identifying people who are undernutrition and require nutritional support during TB treatment [71]. Additionally, future studies are needed to evaluate the time-varying effect of BMI on mortality in people with TB [68]. Also, it is important to consider the different cut-off values for BMI used in different studies, which can make it difficult to compare results [71].

Our study findings highlighted the need to provide systematic BMI assessment among people with TB and tailor the interventions based on various socioeconomic and health-related factors. Integrating nutrition services into TB diagnostic, treatment, and care programs, implementing systematic BMI assessment for all people with TB, training healthcare providers on BMI assessment, and collaborating between National TB and National Nutrition Programs to establish standardized guidelines for BMI assessment and nutritional care in people with TB. Targeted interventions for individuals with undernutrition, young age groups, people with MDR/RR-TB, and people experiencing significant weight loss are also recommended. Additionally, dietary counselling and education should be delivered, especially for young age groups, to encourage a diversified and balanced diet. Regular monitoring and follow-up should be conducted to assess progress and identify areas requiring improvement. Working collaboratively across sectors, engaging with governments, non-governmental organizations, and community stakeholders to implement social protection measures and address socioeconomic determinants of undernutrition is crucial. This involves adopting promotive and transformative approaches to social protection, which help people rise out of poverty, and designing and implementing nutrition-sensitive social protection interventions that target the nutritionally vulnerable. Finally, improving undernutrition among people hospitalized due to TB should be prioritized by immediately assessing nutritional status upon admission, screening for potential causes of malnutrition beyond TB, providing targeted nutritional supplementation and considering extended periods of nutritional support for people with severe undernutrition or those with multiple risk factors. By implementing these recommendations, nutritional outcomes for people with TB in Lao PDR could be improved.

## Limitations

This study had limitations. Firstly, it lacks national representativeness, relying primarily on selected central and provincial hospitals, which may under or overestimate the prevalence of undernutrition among people with TB nationwide. Additionally, this study did not assess the prevalence of undernutrition and financial constrains in rural vs urban population. Some studies

found undernutrition was more prevalent among people with low income, in rural areas, and larger family size [34,72]. However, other study found that the odds of undernutrition among people with TB were 3.84 times higher in urban areas than in rural areas [32]. Moreover, the results pertain solely to people with TB diagnosed at NTP linked to public hospitals, excluding those diagnosed by private practitioners and missing cases. Moreover, this study incorporated participants who had either not initiated TB treatment or had been undergoing therapy for less than a week. This could underestimate the prevalence of undernutrition, primarily because undernutrition may not manifest as severely in the early stages of TB compared to individuals with a more extended duration of the disease [34]. Finally, undernutrition was assessed by exclusive reliance on BMI as the sole metric for evaluating the prevalence of undernutrition, without the inclusion of supplementary or complementary measurements. Further research is needed to provide a more comprehensive understanding of undernutrition prevalence among people with TB in Lao PDR, considering various healthcare settings and populations.

## Conclusion

A high prevalence of undernutrition is consistently observed among people diagnosed with TB at their diagnosis in Lao PDR, underscoring a significant public health challenge. The study findings emphasize the urgent need for systematic nutritional assessment and counselling as integral components of TB care. Addressing undernutrition is crucial not only to improve overall health outcomes for individuals with TB but also to mitigate the heightened risk of TB relapse and mortality. Additionally, undernutrition weakens the immune system, increasing susceptibility to primary TB infection or activation of latent TB, and is closely linked to unfavourable treatment outcomes. Therefore, proactive identification and intervention to address undernutrition in people with TB are imperative steps toward enhancing their overall health outcomes.

## Supporting information

**S1 Table. Univariate and multivariate analysis for identifying risk factors associated with low body mass index at the time of TB diagnosis.**
(DOCX)

## Acknowledgments

We first would like to thank study participants and their household members who consented to participate in this study. Also, we acknowledge the contribution of clinical dieticians, medical doctors, nurses who participated in the training sessions and contributed to data collections in study sites. Special recognition goes to Dr Khamphet Phoumin, Dr Phonenaly Chittamany, for their dedicated assistance in the data collection process. We acknowledge the support provided by Phitsada Siphanthong. We also appreciate the invaluable technical insights provided by Dr Kalpeshsinh Rahevar to enhance the quality of the manuscript. Funding of this study was provided by the WHO Regional Office for the Western Pacific. V.S., F.M., and T.Y. are staff members of WHO. The authors alone are responsible for the views expressed in this publication and they do not necessarily represent the decisions or policies of WHO.

## Author contributions

**Conceptualization:** Donekham Inthavong, Phonesavanh Keonakhone, Nobuyuki Nishikiori, Takuya Yamanaka.

**Data curation:** Hend Elsayed, Takuya Yamanaka.

**Formal analysis:** Hend Elsayed, Takuya Yamanaka.

**Funding acquisition:** Manami Yanagawa, Fukushi Morishita.

**Investigation:** Donekham Inthavong, Phonesavanh Keonakhone, Vilath Seevisay, Somdeth Souksanh, Sakhone Suthepmany, Misouk Chanthavong, Xaysomvang Keodavong, Phonesavanh Kommanivanh, Phitsada Siphanthong.

**Methodology:** Donekham Inthavong, Nobuyuki Nishikiori, Takuya Yamanaka.

**Project administration:** Donekham Inthavong, Vilath Seevisay, Somdeth Souksanh, Sakhone Suthepmany, Phitsada Siphanthong, Manami Yanagawa, Fukushi Morishita.

**Resources:** Donekham Inthavong, Phonesavanh Keonakhone, Misouk Chanthavong, Xaysomvang Keodavong, Phonesavanh Kommanivanh, Phengsy Sengmany, Buahome Sisounon, Manami Yanagawa.

**Software:** Hend Elsayed, Takuya Yamanaka.

**Supervision:** Hend Elsayed, Donekham Inthavong, Phonesavanh Keonakhone, Vilath Seevisay, Somdeth Souksanh, Fukushi Morishita, Takuya Yamanaka.

**Validation:** Hend Elsayed, Somdeth Souksanh, Takuya Yamanaka.

**Visualization:** Hend Elsayed, Takuya Yamanaka.

**Writing – original draft:** Hend Elsayed, Takuya Yamanaka.

**Writing – review & editing:** Jacques Sebert, Fukushi Morishita, Takuya Yamanaka.

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
