## [Decision Letter · Decision Letter 0]

15 Apr 2025

PONE-D-25-02498The prevalence of undernutrition and associated risk factors in people with tuberculosis in Lao People's Democratic RepublicPLOS ONE

Dear Dr. Elsayed,

Thank you for submitting your manuscript to PLOS ONE. After careful consideration, we feel that it has merit but does not fully meet PLOS ONE’s publication criteria as it currently stands. Therefore, we invite you to submit a revised version of the manuscript that addresses the points raised during the review process.

We look forward to receiving your revised manuscript.

Kind regards,

Mohammed Hasen Badeso, Epidemiologist

Academic Editor

PLOS ONE

**Journal Requirements:**

1. When submitting your revision, we need you to address these additional requirements. Please ensure that your manuscript meets PLOS ONE's style requirements, including those for file naming. The PLOS ONE style templates can be found at https://journals.plos.org/plosone/s/file?id=wjVg/PLOSOne_formatting_sample_main_body.pdf and https://journals.plos.org/plosone/s/file?id=ba62/PLOSOne_formatting_sample_title_authors_affiliations.pdf 2. Please update your submission to use the PLOS LaTeX template. The template and more information on our requirements for LaTeX submissions can be found at http://journals.plos.org/plosone/s/latex. 3. Please include a complete copy of PLOS’ questionnaire on inclusivity in global research in your revised manuscript. Our policy for research in this area aims to improve transparency in the reporting of research performed outside of researchers’ own country or community. The policy applies to researchers who have travelled to a different country to conduct research, research with Indigenous populations or their lands, and research on cultural artefacts. The questionnaire can also be requested at the journal’s discretion for any other submissions, even if these conditions are not met.  Please find more information on the policy and a link to download a blank copy of the questionnaire here: https://journals.plos.org/plosone/s/best-practices-in-research-reporting. Please upload a completed version of your questionnaire as Supporting Information when you resubmit your manuscript. 4. Thank you for stating the following financial disclosure: Funding of this study was provided by the World Health Organization (WHO) Regional Office for the Western Pacific (WPRO).   Please state what role the funders took in the study.  If the funders had no role, please state: "The funders had no role in study design, data collection and analysis, decision to publish, or preparation of the manuscript." If this statement is not correct you must amend it as needed. Please include this amended Role of Funder statement in your cover letter; we will change the online submission form on your behalf. 5. Thank you for stating the following in the Acknowledgments Section of your manuscript: We first would like to thank study participants and their household members who consented to participate in this study. Also, we acknowledge the contribution of clinical dieticians, medical doctors, nurses who participated in the training sessions and contributed to data collections in study sites. Special recognition goes to Dr Khamphet Phoumin, Dr Phonenaly Chittamany, for their dedicated assistance in the data collection process. We acknowledge the support provided by Phitsada Siphanthong. We also appreciate the invaluable technical insights provided by Dr Kalpeshsinh Rahevar to enhance the quality of the manuscript. Funding of this study was provided by the WHO Regional Office for the Western Pacific. We note that you have provided funding information that is not currently declared in your Funding Statement. However, funding information should not appear in the Acknowledgments section or other areas of your manuscript. We will only publish funding information present in the Funding Statement section of the online submission form. Please remove any funding-related text from the manuscript and let us know how you would like to update your Funding Statement. Currently, your Funding Statement reads as follows: Funding of this study was provided by the World Health Organization (WHO) Regional Office for the Western Pacific (WPRO).   Please include your amended statements within your cover letter; we will change the online submission form on your behalf. 6. We note that you have indicated that there are restrictions to data sharing for this study. For studies involving human research participant data or other sensitive data, we encourage authors to share de-identified or anonymized data. However, when data cannot be publicly shared for ethical reasons, we allow authors to make their data sets available upon request. For information on unacceptable data access restrictions, please see http://journals.plos.org/plosone/s/data-availability#loc-unacceptable-data-access-restrictions.  Before we proceed with your manuscript, please address the following prompts: a) If there are ethical or legal restrictions on sharing a de-identified data set, please explain them in detail (e.g., data contain potentially identifying or sensitive patient information, data are owned by a third-party organization, etc.) and who has imposed them (e.g., a Research Ethics Committee or Institutional Review Board, etc.). Please also provide contact information for a data access committee, ethics committee, or other institutional body to which data requests may be sent. b) If there are no restrictions, please upload the minimal anonymized data set necessary to replicate your study findings to a stable, public repository and provide us with the relevant URLs, DOIs, or accession numbers. Please see http://www.bmj.com/content/340/bmj.c181.long for guidelines on how to de-identify and prepare clinical data for publication. For a list of recommended repositories, please see https://journals.plos.org/plosone/s/recommended-repositories. You also have the option of uploading the data as Supporting Information files, but we would recommend depositing data directly to a data repository if possible. Please update your Data Availability statement in the submission form accordingly. 7. When completing the data availability statement of the submission form, you indicated that you will make your data available on acceptance. We strongly recommend all authors decide on a data sharing plan before acceptance, as the process can be lengthy and hold up publication timelines. Please note that, though access restrictions are acceptable now, your entire data will need to be made freely accessible if your manuscript is accepted for publication. This policy applies to all data except where public deposition would breach compliance with the protocol approved by your research ethics board. If you are unable to adhere to our open data policy, please kindly revise your statement to explain your reasoning and we will seek the editor's input on an exemption. Please be assured that, once you have provided your new statement, the assessment of your exemption will not hold up the peer review process. 8. PLOS requires an ORCID iD for the corresponding author in Editorial Manager on papers submitted after December 6th, 2016. Please ensure that you have an ORCID iD and that it is validated in Editorial Manager. To do this, go to ‘Update my Information’ (in the upper left-hand corner of the main menu), and click on the Fetch/Validate link next to the ORCID field. This will take you to the ORCID site and allow you to create a new iD or authenticate a pre-existing iD in Editorial Manager. 9. We notice that your supplementary table is included in the manuscript file. Please remove and upload them with the file type 'Supporting Information'. Please ensure that each Supporting Information file has a legend listed in the manuscript after the references list.

Reviewers' comments:

Reviewer's Responses to Questions

**Comments to the Author**

1. Is the manuscript technically sound, and do the data support the conclusions?

Reviewer #1: No

Reviewer #2: Yes

2. Has the statistical analysis been performed appropriately and rigorously? 

Reviewer #1: No

Reviewer #2: Yes

3. Have the authors made all data underlying the findings in their manuscript fully available?

Reviewer #1: Yes

Reviewer #2: Yes

4. Is the manuscript presented in an intelligible fashion and written in standard English?

Reviewer #1: Yes

Reviewer #2: Yes

5. Review Comments to the Author

**Reviewer #1:**  The reported odds ratios and their corresponding log(odds ratios) are inconsistent and require re-analysis. Ensure that computations align correctly before finalizing the results.

(Figure 1):

o The odds ratio (OR) quantifies the association between an exposure and an outcome. The null hypothesis (no association) corresponds to an OR of 1, meaning the odds of the outcome are the same in both groups.

o Since the logarithm of 1 is 0 (i.e., log(1) = 0), the null value for the log odds ratio (log(OR)) is also 0.

o Your figure shows that all variables have Log(OR) > 0, suggesting that all are positively associated with the outcome. However, this contradicts the textual interpretation of the results using odds ratios.

o Additionally, the antilog of the reported log(OR) values does not match the stated odds ratios. For example, the antilog of 3.1 does not yield an OR of 6.8 for the age group 15–24. A re-analysis is necessary.

**Reviewer #2: ** This study investigates malnutrition among tuberculosis patients at the time of diagnosis in Lao PDR. The results and discussion are presented based on data obtained from central and provincial hospitals in relation to the stated research objectives. Although the representativeness of the collected data in tuberculosis patients is a concern, the study appropriately discusses its limitations. Given its valuable insights into the topic, this manuscript is considered worthy of acceptance.

6. PLOS authors have the option to publish the peer review history of their article (what does this mean? ). If published, this will include your full peer review and any attached files.

**Do you want your identity to be public for this peer review?** For information about this choice, including consent withdrawal, please see our Privacy Policy .

Reviewer #1: No

Reviewer #2: No

---

## [Author Response · Author response to Decision Letter 1]

24 Apr 2025

Thank you for your further review and comments. We added our responses to each comment with red font text and revised the manuscript with track changes accordingly.

Reviewer #1: The reported odds ratios and their corresponding log(odds ratios) are inconsistent and require re-analysis. Ensure that computations align correctly before finalizing the results.

(Figure 1):

o The odds ratio (OR) quantifies the association between an exposure and an outcome. The null hypothesis (no association) corresponds to an OR of 1, meaning the odds of the outcome are the same in both groups.

o Since the logarithm of 1 is 0 (i.e., log(1) = 0), the null value for the log odds ratio (log(OR)) is also 0.

o Your figure shows that all variables have Log(OR) > 0, suggesting that all are positively associated with the outcome. However, this contradicts the textual interpretation of the results using odds ratios.

Thank you for your clarifications. The results are estimated with antilog scale and therefore, for all the results shown in fig 1, the null hypothesis of the analysis still corresponds to an OR of 1, not zero. Given that a variable (weight decreased) has much a wider confidence interval (i.e. CI 2.3-36.4 for hospitalization), we applied a log scale to the x-axis label (not results themselves) only for the visualization purposes. Hence, we do not change the analytical approach.

We understand that there might be confusions as raised by the reviewer, and therefore we revised the axis title and also added a footnote for this matter.

o Additionally, the antilog of the reported log(OR) values does not match the stated odds ratios. For example, the antilog of 3.1 does not yield an OR of 6.8 for the age group 15–24. A re-analysis is necessary.

Thank you for spotting these inconsistencies. We revised the paragraph in result section accordingly.

… in Figure 1 (results with univariate and multivariate analysis are available in Table S1 in appendix). There were substantially heightened risk of low BMI among different age groups. Specifically, individuals aged 15-24 years old exhibited a remarkably higher risk (AOR: 6.9, 95%CI: 2.2-23.2) compared to those aged 35-44 years old. Similarly, the risk remained elevated for those aged 25-34 years old (AOR: 2.2, 95%CI: 0.9-5.5) and even for individuals aged 65 years and above (AOR=2.0, 95% CI: 0.8-4.8). The presence of drug resistance also emerged as a significant risk factor for low BMI (AOR: 3.2, 95% CI=1.0-11.8). Moreover, patients who were hospitalized until the diagnosis of TB faced a substantially higher risk of low BMI (AOR: 3.4, 95% CI=2.0-5.9). This indicates that the severity of the TB condition, requiring hospitalization, is associated with a greater likelihood of low BMI at the time of TB diagnosis. Notably, individuals who experienced weight loss in the last 6 months prior to TB diagnosis were at a significantly elevated risk of having a low BMI (AOR: 7.8, 95% CI: 2.3-36.4). Finally, patients who were categorized as being below the international poverty line at the time of TB diagnosis also faced an increased risk of low BMI (AOR: 1.9, 95% CI 1.0-3.6).

*x-axis is with log-scale.

Reviewer #2: This study investigates malnutrition among tuberculosis patients at the time of diagnosis in Lao PDR. The results and discussion are presented based on data obtained from central and provincial hospitals in relation to the stated research objectives. Although the representativeness of the collected data in tuberculosis patients is a concern, the study appropriately discusses its limitations. Given its valuable insights into the topic, this manuscript is considered worthy of acceptance.

Thanks for your favourable comment and decision to our manuscript.

---

## [Editor Report · Decision Letter 1]

2 May 2025

The prevalence of undernutrition and associated risk factors in people with tuberculosis in Lao People's Democratic Republic

PONE-D-25-02498R1

Dear Author(s),

We’re pleased to inform you that your manuscript has been judged scientifically suitable for publication and will be formally accepted for publication once it meets all outstanding technical requirements.

Kind regards,

Mohammed Hasen Badeso, Epidemiologist

Academic Editor

PLOS ONE
---

## [Editor Report · Acceptance letter]

PONE-D-25-02498R1

PLOS ONE

Dear Dr. Elsayed,

I'm pleased to inform you that your manuscript has been deemed suitable for publication in PLOS ONE. Congratulations! Your manuscript is now being handed over to our production team.

Kind regards,

on behalf of

Mr Mohammed Hasen Badeso

Academic Editor

PLOS ONE